# Satisfaction and attrition in the UK healthcare sector over the past decade

**Neel Ocean** [1]*, **Caroline Meyer**[2]

**1** WMG, University of Warwick, Coventry, United Kingdom, **2** University of Warwick, Coventry, United Kingdom

* neel.ocean@warwick.ac.uk

**Data Availability Statement:** NHS data are available from: https://digital.nhs.uk/data-and-information/publications/statistical/nhs-workforce-statistics UKHLS data is available from https://beta.ukdataservice.ac.uk/datacatalogue/studies/study?id=6614.

## Abstract

Existing literature has highlighted concerns over working conditions in the UK National Health Service (NHS), with healthcare workers frequently citing work-life balance issues and stress as being drivers of attrition and burnout. However, we do not know whether these problems have become *worse* over time, particularly over the past decade, during which there have been multiple shocks to the UK healthcare system. To investigate this, we analysed data from NHS monthly workforce statistics and the UK Household Longitudinal Study. Three times as many workers left the NHS in 2021 for work-life balance reasons than in 2011, while estimated satisfaction with one's amount of leisure time for healthcare workers fell by three times the amount that it fell for non-healthcare workers. Both satisfaction with amount of leisure time and satisfaction with income have remained lower for healthcare workers than for other public sector workers. By 2020, a worker that had low satisfaction with their amount of leisure time was as much as 22 percentage points less likely than in 2010 to remain in healthcare in the following year. Overall, working conditions in UK healthcare have deteriorated between 2010 and 2020, especially relative to the private sector. However, overall job satisfaction has fallen faster in other areas of the public sector than it has in healthcare, which may indicate wider issues within the UK public sector as a whole.

## 1 Introduction

Since 2010, the UK healthcare system has been subject to a range of both domestic and global shocks. Globally, the COVID-19 pandemic led to a reconfiguration of services and an improvisation of resources in order to cope with demand [1]. In the UK, large numbers of people volunteered to provide extra support such as delivering supplies or engaging in activities to mitigate negative effects from social isolation [2, 3]. National Health Service (NHS) staff stretched themselves to meet demand, and former employees temporarily returned to relieve resourcing constraints. This extraordinary shock to the labour market led many to re-evaluate their work-life balance and career choices. In what is now known as the 'Great Resignation' or 'Big Quit' [4, 5], a large wave of workers across various industries in developed economies voluntarily left their jobs (e.g. [6]). The pandemic has led to employees becoming more demanding of their employment conditions, e.g. more flexible working [5]. NHS workers are

**Funding:** The author(s) received no specific funding for this work.

**Competing interests:** The authors have declared that no competing interests exist.

no exception, with burnout often being tied to plans to leave or to reduce working hours [7]. Many NHS staff have been complaining of increased bureaucracy and an increasing workload, and this appears to have been amplified by the pandemic [8].

However, the NHS has also had other shocks over the past decade. Most notably, the UK government in 2012 introduced the Health and Social Care Act, which represented a significant structural change to the NHS, mostly in England [9, 10]. This was primarily designed to decentralise healthcare provision by eliminating primary care trusts, instead creating clinical commissioning groups (CCGs) comprised of GPs that were responsible for commissioning primary care services for their local areas [11]. The bill was not without controversy. Some argued that the reforms took the NHS closer towards a privatised healthcare system because they were designed to introduce more market-based competition [12, 13].

Given these shocks, it would be natural to ask how healthcare worker satisfaction has evolved in the UK during this period. However, work in this area to date has generally been qualitative or has involved small samples at fixed points in time. Hence, there is scant existing research that can provide an answer. Therefore, the main research aim of this paper is to provide quantitative evidence in order to answer the following questions: (i) *have working conditions for UK healthcare workers deteriorated on aggregate over the past decade*?; and (ii) *have conditions for UK healthcare workers deteriorated relative to workers outside the sector over the past decade*? In the present study, we tracked the reasons for leaving the NHS over the past decade to show that work-life balance issues have become increasingly responsible for job attrition in the NHS. We then used individual-level longitudinal data from the nationally representative Understanding Society survey in order to understand how satisfaction levels have changed over time for healthcare workers, both absolutely and in relative terms when compared to workers outside the sector. Finally, we used these data to uncover the degree to which satisfaction levels can predict attrition in the healthcare sector.

## 1.1 Background literature

We next review the existing literature on NHS working conditions. We focus on two broad topic areas. First, we review evidence on job attrition in healthcare, because changes in leaving behaviour are an important signal of changes in working conditions. Second, we review evidence on job-related satisfaction or conditions that existing healthcare workers are having to deal with.

**1.1.1 Job attrition in the NHS.** The majority of existing studies that have attempted to investigate the reasons behind NHS workers leaving have been limited to smaller samples, where the outcomes for a specific occupation was the main focus rather than for the entire sector (such as for nursing [14]). A number of these studies have been qualitative and involved interviews [15–18] or other combinations of qualitative survey and narrative analysis [19–25]. While insightful, the majority of these studies have sample sizes well below 100, and so it is difficult to draw any general conclusions about the reasons behind attrition, particularly if there is a sampling or response bias.

Of the aforementioned studies, only three had sample sizes of more than 1000 [22, 23, 25]. Loan-Clarke et al. [23] studied 'stayers', 'leavers', and 'returners' within the NHS at two time points: 2005 (n = 1925) and 2007 (n = 719). They found that the main reason for staying in the NHS at t = 2 was general satisfaction with the work itself, whereas the main reasons for leaving the NHS were related to stress and workload, poor management in the NHS, and a want or need to look after children. They also found that the main change staff had noticed between t = 1 and t = 2 was an increase in redundancies and posts remaining unfilled as a cost-cutting measure. This naturally leads to further problems with workload and stress for the remaining

employees. Lambert et al. [22] surveyed UK medical graduates three years post-graduation (n = 5291), and found that 59% of respondents 'were not definitely intent on remaining in UK medicine'. Multiple reasons were cited for this. However, complaints primarily related to the state of the NHS, pay, work-life balance, and working conditions were more common in 2015 than in 2011. These were also the primary reasons given by doctors for leaving medicine completely. Junior doctors that had gone overseas felt more valued, with better levels of pay and staff coverage, as well as more choice on location and rotas [26]. Ryan et al. [25] studied counsellors in the UK in 2017 (n = 1918), and a similar pattern of discontent emerged. The main themes highlighted were: 'squeezed' (out of the NHS), 'underpaid', 'pressurised', and 'undervalued'. In particular, the added pressure and workload not only worsened staff well-being but had a subsequent negative impact on patients.

High quality quantitative studies on NHS job attrition are less common. Many of them suffer from small samples or lack adequate regression modelling to robustly estimate relationships [14, 27–31], though some larger sample regression-based studies do exist [32–36]. For registered nurses, work pressure and work-related stress emerged from structural equation modelling (n = 16,707) as the strongest positive correlates with intent to leave the NHS (other than organisation type) [32]. Aggression from both colleagues and patients is significantly related to the intent to leave [33], though it is not clear from the study how important this reason is relative to others. Hann et al. [36] and Fletcher et al. [34] both study GP survey data in samples >1000. Extreme job dissatisfaction is strongly related to leaving practice, as would be expected, though high job satisfaction did not necessarily reduce the likelihood of leaving [36]. 21.7% of 2177 GPs said they were likely to permanently leave patient care in the next two years, and 54.9% reported having either low or very low morale [34].

**1.1.2 NHS job satisfaction and working conditions.** Given that job attrition is closely related to working conditions, it is also sensible to review existing literature on job satisfaction and working conditions within the NHS. The majority of studies we found on job satisfaction focused on nurses [8, 37–40], with some looking at specialist doctors [41, 42] and GPs [43]. Poor working conditions (i.e. structural problems within the NHS such as poor management or poor facilities) as well as overwork/stress and dissatisfaction with pay were common negative correlates with job satisfaction, as they were with job attrition. Unsurprisingly, the COVID-19 pandemic appeared to act as a negative external shock to job satisfaction for healthcare workers in the NHS [44]. However, this study was cross-sectional and so their conclusion hinged on remembered experiences that could be subject to bias (see [45]).

For comparison, there are very few recent UK-based studies that measure job satisfaction in sectors outside of healthcare. One area that has been looked at, possibly due to the relative ease of data collection, is academia. Being in a cooperative environment is related to higher job satisfaction among academic economists [46], and the perceived strength of a departmental chair is positively related to job satisfaction while being negatively related to leaving intentions, which may also be related to their impact on fostering a positive academic environment [47]. While academia and healthcare jobs perhaps have little in common, it seems as though having positive relationships with both managers and colleagues may be uniquely important for job satisfaction and employee retention more generally.

The majority of studies we found for working conditions within the NHS were qualitative and involved relatively small samples, as was the case for job attrition. GPs appear to be less and less able to cope with an ever increasing workload, combined with increased bureaucratic guidelines such as the Quality and Outcomes Framework [48]. This has negative spillovers on leisure time, with 70% of doctors in one study reporting difficulty in personal relationships and 87% reporting an adverse impact on hobbies due to work schedules [49]. The same study found that only 26% of UK doctors (n = 417) were satisfied with their work-life balance [49],

with common reasons being related to difficult schedules and having to work far from home. In a 2014 sample of doctors who graduated in 1974 or 1977 (n = 3695), 44.4% stated that working as a doctor has had adverse effects on their health or well-being and 42.6% said that the NHS was not a good employer when doctors became ill themselves [50]. Research on trainees in specialist areas of medicine suggests that stress and poor work-life balance were common concerns in terms of future career prospects [51, 52], and junior doctors may avoid certain specialities because of this [53]. 58% of 168 emergency department workers surveyed at one UK hospital were dissatisfied with their work-life balance, and 42% felt burnt out while 74% felt that they would be at a high risk of burnout within the next 6 months [54]. Nurses in particular appear to suffer from poor work-life balance where a gruelling schedule typically spills over into a reduced capacity to enjoy life outside of work [54, 55]. This is exacerbated by a belief in some areas that staff in such a profession are not entitled to a personal life outside of work [56].

Quantitative studies on working conditions in the NHS paint a similar picture to the issues raised by more qualitative studies. Again, large-scale samples are relatively rare, with many studies involving small convenience samples of (usually) workers in specific areas of healthcare [57–59]. The more robust quantitative studies we found still supported a generally negative evaluation of working conditions in the NHS. For example, a study that used the GHQ-12 to measure NHS staff well-being (n = 3606) with a logistic regression found a significant relationship between higher work-life conflict and worse mental well-being [60]. In obstetrics and gynaecology professionals, 36% met the criteria for burnout according to the Maslach Burnout Inventory (n = 3073), and this proportion was significantly higher for trainees [61]. Burnout was found to significantly increase the likelihood of depression, anxiety, alcohol/drug abuse, sleep issues, binge eating, and suicidal thoughts [61, 62].

In summary, it is clear that working conditions within the NHS do not seem favourable, and structural changes appear to be required. However, if conditions have been poor for some time, then responses to surveys of this nature will predominantly be negative, especially if there is selection bias in the workers that choose to participate. Therefore, we think the more interesting question is: *have working conditions in the NHS worsened over time*? One shortcoming of existing studies is that they do not study longitudinal data or patterns, which makes it difficult to understand how conditions have evolved over time, or whether there are any intertemporal predictors of attrition. Another major shortcoming is that most quantitative studies generally fail to capture individual-level covariates that are known to affect job satisfaction, such as working hours, education, age, income, and marital status (e.g. see [63–65]). It is therefore difficult to determine from existing literature whether working conditions in UK healthcare have deteriorated over time, both in absolute terms and relative to other sectors. The present study addresses this shortcoming by utilising two national datasets that span the previous decade. We are interested in this period in particular because, as we previously discussed, two significant shocks to the NHS occurred during this time. The 2012 Health and Social Care Act attempted to shift care away from specialists in hospitals towards local GPs and cheaper secondary care services [11], while the COVID-19 pandemic placed considerable strain on the NHS (as well as public health services globally) [66, 67].

## 2 Materials and methods

The main question we aim to answer in this study is: have working conditions in UK healthcare deteriorated over the past decade? We tackle this question using two datasets, which we provide more details of next. All statistical analyses were performed using Stata software [68].

## 2.1 NHS workforce statistics

To provide summary information about the reasons behind NHS job attrition, we use publicly available NHS workforce statistics [69]. This family of datasets track headcounts across all departments and geographical regions of the NHS. These data are officially collected by the NHS and have been released monthly on the NHS's website starting from September 2013. Data are from English NHS organisations that use the Electronic Staff Record (ESR), and include hospital and community health service workers that are paid directly by an NHS organisation (i.e. an NHS Trust or a Clinical Commissioning Group). This means that the data do not include GPs and dentists, for example, who are given payments for their practice based on the number of registered patients as well as their performance. We focus on a time series of the reasons for leaving the NHS (or staff movements within the NHS) that spans from Q1 2011–2012 to Q3 2021–2022. The NHS quarters run from Apr-Jun, Jul-Sep, Oct-Dec, Jan-March for Q1, Q2, Q3, Q4 respectively. The dataset includes 39 possible reasons for leaving an NHS job, which include involuntary reasons (e.g. a fixed-term contract that has come to an end), as well as various voluntary reasons. We are limited in how we can analyse these data, given that they are not at the individual level. However, the recency of the data and long timeframe covered allow us to see how working conditions have evolved within the NHS over the past decade.

We calculated how the proportion of workers leaving the service for work-life reasons has changed over this period on average using time series smoothing methods. A four-period moving average was calculated in order to smooth quarterly data over seasonal fluctuations:

$$\overline{x_t} = \frac{x_t + x_{t-1} + x_{t-2} + x_{t-3}}{4}$$

where $x_t$ is the number of people leaving the NHS at time $t$. We also estimated autoregressive models of the following general form on the smoothed data in some analyses to formally test for stationarity:

$$\overline{x_t} = \beta_1 + \beta_2 \overline{x_{t-1}} + \epsilon_t$$

## 2.2 UK Household Longitudinal Survey ("Understanding Society")

To investigate the individual level factors that may be predictive of attrition, and lower levels of job-related satisfaction and subjective well-being, we use data from the UK Household Longitudinal Survey (UKHLS), also known as "Understanding Society" [70]. This data source has been used recently for many studies on subjective well-being and mental health (e.g. [71–74]), as well as general job satisfaction (e.g. [75]). However, to the best of our knowledge, it has not been used to investigate healthcare worker satisfaction or its likely impact on job attrition. The dataset consists of 11 waves, and covers a year range between 2009 and 2021, which largely overlaps with the data available from the NHS workforce statistics. The survey follows approximately 50,000 individuals, though there is replacement throughout the collection period, and not all individuals answer every question. Therefore, the precise number of individuals and observations for a particular analysis will vary depending on the specific waves and variables included.

In order to identify healthcare workers, we used the variable *jbsectpub*. This lists the type of public sector work the individual is involved in, if they were employed in the public sector at the time of survey completion. One of the response options for this question was 'A health authority or NHS trust'. We labelled individuals as being 'in healthcare' for a given wave if they stated that they were employed in a health authority or NHS trust in that wave. On

average, healthcare workers comprised 6.57% of workers in the sample, with an average of 1551 healthcare workers per wave. The proportion of males over the study period was approximately 52.1% outside of healthcare, and 18.8% in healthcare. For each wave of the study, those that were unemployed were excluded from all analyses.

Aside from providing standard demographic and economic information for each participant, such as age, marital status, education, income, and so on, the survey also contains data on various measures of satisfaction. We utilised four different satisfaction measures: overall job satisfaction, satisfaction with the amount of leisure time one has, income satisfaction, and satisfaction with one's health. Satisfaction with the amount of leisure time was of particular interest to us, given that poor work-life balance is a common complaint in existing literature on NHS workers. These variables were measured on a 7-point scale, where 1 means 'completely dissatisfied' and 7 means 'completely satisfied'. We also looked at six different job-related mood variables, though these were only collected in even-numbered waves (i.e. waves 2, 4, 6, 8, 10). The wording of these questions was "Thinking of the past few weeks, how much of the time has your job made you feel. . .", followed by one of six different negative feelings. These items were measured on a 5-point frequency scale where 1 means 'never' and 5 means 'all of the time'.

**2.2.1 Fixed-effects analysis specification.** We estimated a number of fixed-effects (FE) regressions, which make within-person comparisons between time-varying variables possible due to the longitudinal nature of the data. The general form of the models we estimate is:

$$y_{i,t} = \alpha + \boldsymbol{\beta} X_{i,t} + \boldsymbol{\gamma} Z_{i,t-1} + \boldsymbol{\psi} t X_{i,t} + \boldsymbol{\phi} t Z_{i,t-1} + \theta t + \delta_i + \epsilon_{i,t}$$

where $y_{i,t}$ is the dependent variable for individual $i$ at time $t$, $\alpha$ is a constant, $X_{i,t}$ is a matrix of contemporaneous variables, $Z_{i,t-1}$ is a matrix of lagged variables, $\delta_i$ is the unobserved individual-level fixed effect, $t$ is a time trend (i.e. the survey wave), and $\epsilon_{i,t}$ is the error term. In all regressions, we include current age, age-squared, marital status, number of dependent children, income, and whether the individual is self-employed or not. For some regressions, we also include lagged variables for various measures of satisfaction in order to try to understand whether specific kinds of dissatisfaction make it more likely that individuals will leave healthcare, or will have worse overall levels of satisfaction (both in terms of one's job and also in terms of overall life satisfaction). The key parameters of interest are $\psi$ or $\phi$ (depending on the precise specification estimated). These are interaction terms between time and our independent variables of interest, so that we can estimate how the relationship between the independent variable and dependent variable changes over time.

## 3 Results and analysis

### 3.1 Reasons for leaving NHS jobs

We first look at data from NHS workforce statistics to study patterns of leaving behaviour over time. In Fig 1a, we plot the headline number of leavers for every quarter from Q1 2011 to Q3 2021, as well as the moving average calculated as described in section 2.1. For simplicity, we henceforth only refer to the calendar year attached to the first quarter even though the NHS administrative year spans two calendar years. Because the smoothing procedure uses only lags, the graph is only fully smoothed from Q1 2012 onwards. Apart from two shocks, one in 2012 and another in 2020, the number of leavers has remained relatively constant. We can see this stationarity more robustly by estimating an autoregressive model on the smoothed data. Estimating this model for total leavers gives $\widehat{\beta_2} = 0.268$, with 95% confidence interval [-0.014, 0.550], n = 42. Alternatively, estimating the first difference yields 240.69 with 95% confidence interval

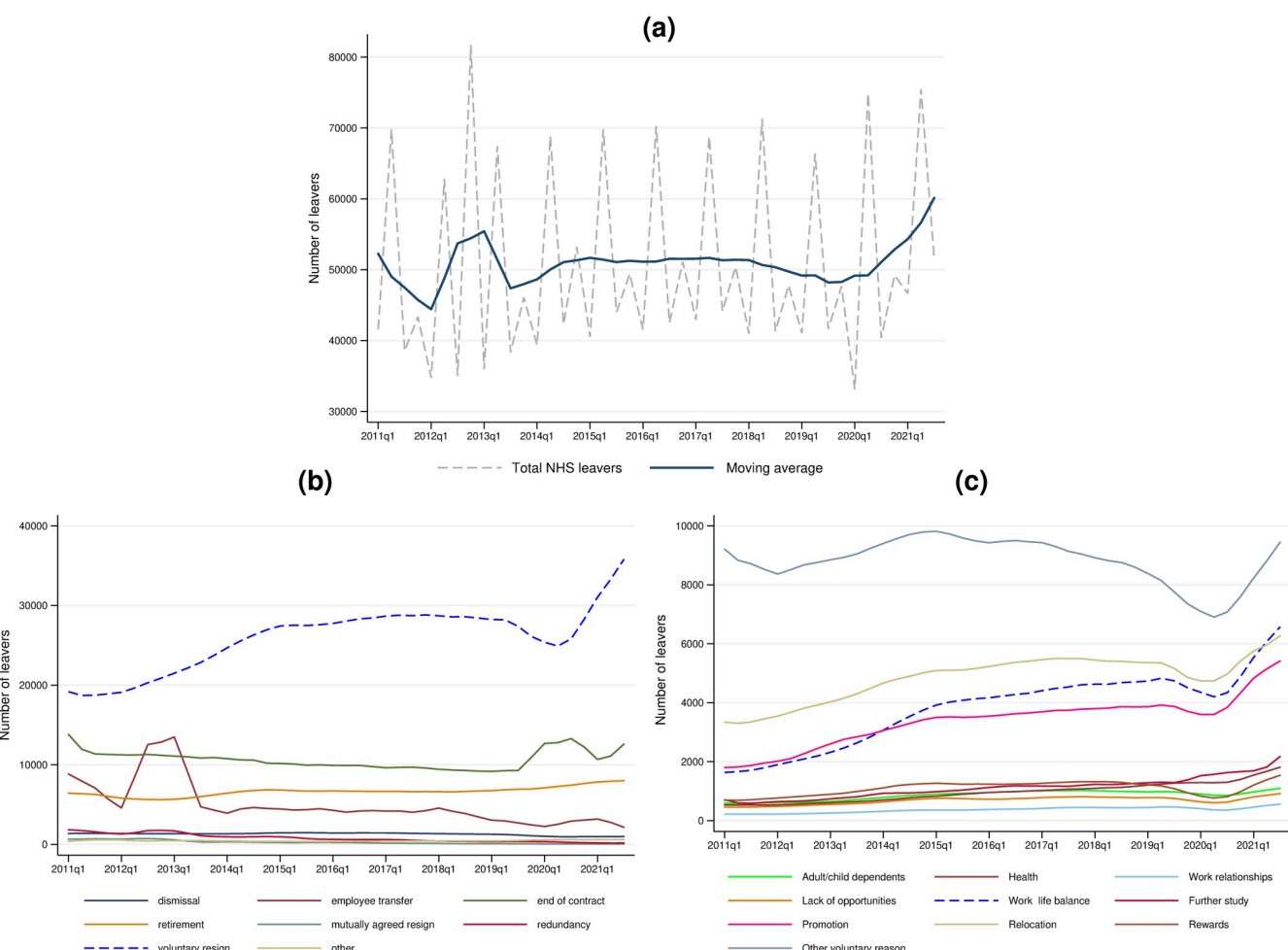

**Fig 1. Number of people leaving an NHS job from Q1 2011 to Q3 2021, using NHS workforce statistics.** (a) Total NHS job leavers. (b) Breakdown of main reasons for leaving. (c) Breakdown of reasons for leaving voluntarily.

[-683.31, 1164.69], and an AR(1) error term of -0.325 [-0.692, 0.424]. An augmented Dickey-Fuller test with the null hypothesis that there is a unit root ($H_0$: $\beta_2 = 1$) gives $p < 0.00005$. Therefore, it appears that the time series is stationary, and so any variation in the total number of leavers is highly likely to represent a random shock. As mentioned in the introduction, the most likely explanations for the two shocks are the introduction of the 2012 Health and Social Care Act leading to a restructuring of contracts, and then the COVID-19 pandemic from 2020.

Fig 1b plots the smoothed reasons for leaving within this time period. While the original data contains 39 separate reasons for leaving, we have consolidated them into 8 categories to provide a clearer graph. The consolidated categories are: dismissal, employee transfer, end of a fixed-term contract, retirement (for any reason), mutually agreed resignation, redundancy (both voluntary and compulsory), voluntary resignation, and other (i.e. reasons that are not classifiable under the previous headings). It is clear from Fig 1b that voluntary resignation not only represents the largest proportion of NHS leavers, but also that the proportion of voluntary leavers is increasing over time. Combined non-voluntary reasons account for more leavers than voluntary reasons until Q4 2013. After this, more leavers are voluntary than non-voluntary, accounting for approximately 60% of all leavers towards the end of the study period.

Fig 1c splits voluntary resignations further. For the majority of voluntary resignations, the reason for leaving was not specified in the dataset. These 'other' voluntary leavers do not appear to be increasing in number. However, we see upward trends in the number of people leaving for promotion, relocation, or work-life balance. Augmented Dickey-Fuller tests yield p-values of 0.8976, 0.1458, and 0.9302 for promotion, relocation, and work-life balance respectively. Therefore, all three are non-stationary and likely have an increasing trend, particularly for promotion and work-life balance. As promoted individuals are unlikely to leave the NHS outright, it is the strong increasing trend in *work-life balance leavers* that appears to be driving much of the increase in voluntary resignations. In fact, by the final period within these data, averaged work-life balance becomes the most frequently specified reason for voluntary resignation for the first time. Using smoothed data, leaving for poor work-life balance represented 3.66% of total leavers in Q1 2011, but represented 10.53% of total leavers in Q3 2021—almost triple. A first difference AR(1) model estimate suggests that the number of work-life balance related leavers is increasing by 141.68 people each quarter ($p = 0.144$, with an AR(1) coefficient of 0.775). This implies that over the past decade, 567 additional workers leave NHS jobs every year because of work-life balance reasons. Overall, it appears that there is an increasing trend of NHS workers leaving voluntarily, and increasingly for work-life balance reasons.

## 3.2 Absolute and relative satisfaction of healthcare workers over time

We now analyse longitudinal data from the UKHLS in order to determine healthcare worker satisfaction at the individual level. First, we fit a model to estimate whether healthcare worker satisfaction has decreased over time, and how this change looks relative to non-healthcare worker satisfaction. In terms of the regression specification outlined in section 2.2.1, we are particularly interested in the estimate of $\psi$ because this will tell us whether any change in satisfaction over time is statistically significant. We use four different measures of satisfaction as dependent variables: job satisfaction, satisfaction with the amount of leisure time, income satisfaction, and health satisfaction. For our set of contemporaneous covariates (i.e. $X_{i,t}$ in our specification), we included the standard demographic variables: age (including a quadratic term), marital status, number of dependent children, whether they had a degree, monthly income, and whether the individual was self-employed. No lagged variables were included in these specifications. Our key independent variable was a dummy variable for whether the individual was working in healthcare in that wave. We also interacted this variable with the survey wave. The coefficient of this interaction term provides us with our estimate of $\psi$.

The first four regressions in Table 1 each estimate a different form of satisfaction. Most notably, Regression (2) suggests that there is a significant negative interaction between wave and 'in healthcare' when estimating satisfaction with one's amount of leisure ($p = 0.031$ when using regular OLS standard errors, $p = 0.058$ when using standard errors clustered by individual). This suggests that satisfaction with the amount of leisure time for healthcare workers has, over time, fallen below workers outside of healthcare. This can be seen in the top-right plot shown in Fig 2. Fig 2 plots the moderating effect of being in different sectors on satisfaction over time. Estimated levels of satisfaction with the amount of leisure time fell by 0.04 (from 4.48 to 4.44) on a 1 to 7 scale between waves 1 and 11 for those *not* working in healthcare. For those in healthcare, however, the drop was 0.12 (from 4.51 to 4.39). This is *three times* the drop in satisfaction experienced for workers outside of the healthcare sector. Estimated satisfaction with leisure time for healthcare workers was higher than for non-healthcare workers in the early 2010s, but there appears to have been a 'crossing point' at some time during the middle of the previous decade due to the steeper decline for healthcare workers.

**Table 1. Fixed-effects OLS regressions showing satisfaction over time for healthcare workers, relative to workers outside healthcare.**

| | (1) Job satisfaction | (2) Sat w/ amount leisure time | (3) Income satisfaction | (4) Health satisfaction | (5) Job satisfaction | (6) Sat w/ amount leisure time | (7) Income satisfaction | (8) Health satisfaction |
|---|---|---|---|---|---|---|---|---|
| Wave | 0.00278 | -0.00365 | -0.0122 | -0.0275*** | 0.00925 | -0.000414 | -0.00808 | -0.0257** |
| | (0.00797) | (0.00971) | (0.00963) | (0.0106) | (0.00798) | (0.00973) | (0.00965) | (0.0106) |
| In healthcare | 0.156*** | 0.0462 | 0.0204 | 0.0499 | | | | |
| | (0.0309) | (0.0351) | (0.0347) | (0.0384) | | | | |
| Wave x in healthcare | -0.00468 | -0.00858** | -0.00473 | -0.00641 | | | | |
| | (0.00351) | (0.00398) | (0.00394) | (0.00435) | | | | |
| Not public sector | | | | | -0.238*** | -0.0790** | -0.0671* | -0.0717* |
| | | | | | (0.0315) | (0.0357) | (0.0354) | (0.0391) |
| Wave x not public sector | | | | | 0.0117*** | 0.0121*** | 0.00928** | 0.00837* |
| | | | | | (0.00357) | (0.00404) | (0.00400) | (0.00442) |
| In public sector (excl healthcare) | | | | | -0.00123 | 0.021 | 0.0732** | -0.00684 |
| | | | | | (0.0329) | (0.0374) | (0.0371) | (0.0410) |
| Wave x in public sector (excl healthcare) | | | | | -0.0119*** | -0.000143 | -0.00635 | 0.00165 |
| | | | | | (0.00384) | (0.00435) | (0.00431) | (0.00476) |
| Age | -0.0106 | -0.0313*** | -0.00154 | 0.00747 | -0.012 | -0.0318*** | -0.00229 | 0.00711 |
| | (0.00822) | (0.0100) | (0.00992) | (0.0110) | (0.00821) | (0.0100) | (0.00992) | (0.0110) |
| Age^2 | 0.000170*** | 0.000673*** | 0.000526*** | 0.000171*** | 0.000190*** | 0.000682*** | 0.000538*** | 0.000177*** |
| | (4.43e-05) | (4.94e-05) | (4.89e-05) | (5.40e-05) | (4.43e-05) | (4.94e-05) | (4.89e-05) | (5.40e-05) |
| Married | -0.0116 | -0.0164 | 0.0794*** | -0.00371 | -0.012 | -0.0165 | 0.0793*** | -0.00377 |
| | (0.0138) | (0.0155) | (0.0153) | (0.0169) | (0.0138) | (0.0155) | (0.0153) | (0.0169) |
| # of dep children | 0.00387 | -0.0878*** | -0.0633*** | -0.0154** | 0.00162 | -0.0889*** | -0.0647*** | -0.0160** |
| | (0.00617) | (0.00696) | (0.00690) | (0.00762) | (0.00617) | (0.00697) | (0.00690) | (0.00762) |
| Has degree | -0.0594* | -0.0237 | 0.0281 | 0.0563 | -0.0583* | -0.0217 | 0.0301 | 0.057 |
| | (0.0345) | (0.0385) | (0.0382) | (0.0421) | (0.0345) | (0.0385) | (0.0382) | (0.0422) |
| Monthly income | 2.08e-05*** | -1.90e-05*** | 8.48e-05*** | 1.15e-05*** | 2.02e-05*** | -1.92e-05*** | 8.44e-05*** | 1.13e-05*** |
| | (2.76e-06) | (3.01e-06) | (2.98e-06) | (3.29e-06) | (2.76e-06) | (3.01e-06) | (2.98e-06) | (3.29e-06) |
| Not self employed | -0.239*** | -0.138*** | 0.0428*** | -0.0253 | -0.250*** | -0.141*** | 0.0381** | -0.028 |
| | (0.0145) | (0.0162) | (0.0161) | (0.0177) | (0.0146) | (0.0163) | (0.0162) | (0.0178) |
| Constant | 5.615*** | 4.713*** | 3.473*** | 4.427*** | 5.820*** | 4.774*** | 3.516*** | 4.489*** |
| | (0.293) | (0.358) | (0.355) | (0.392) | (0.295) | (0.360) | (0.357) | (0.394) |
| Observations | 235208 | 224930 | 224880 | 224969 | 235208 | 224930 | 224880 | 224969 |
| R-squared | 0.002 | 0.007 | 0.014 | <0.0005 | 0.003 | 0.007 | 0.014 | <0.0005 |
| # of individuals | 50357 | 48540 | 48522 | 48544 | 50357 | 48540 | 48522 | 48544 |

Note: OLS standard errors are shown in parentheses.

*** $p < 0.01$,

** $p < 0.05$,

* $p < 0.1$

Regression (1) in Table 1 shows that estimated overall job satisfaction has remained significantly higher for healthcare workers relative to non-healthcare workers over the entire period. This may seem surprising given the results from the previous analyses. However, it is consistent with the fact that healthcare workers are often in the sector due to high levels of intrinsic motivation, and it is likely that those that have chosen not to leave the sector have remained for these intrinsic benefits (e.g. from a sense of social responsibility or caring for others). Fig 2

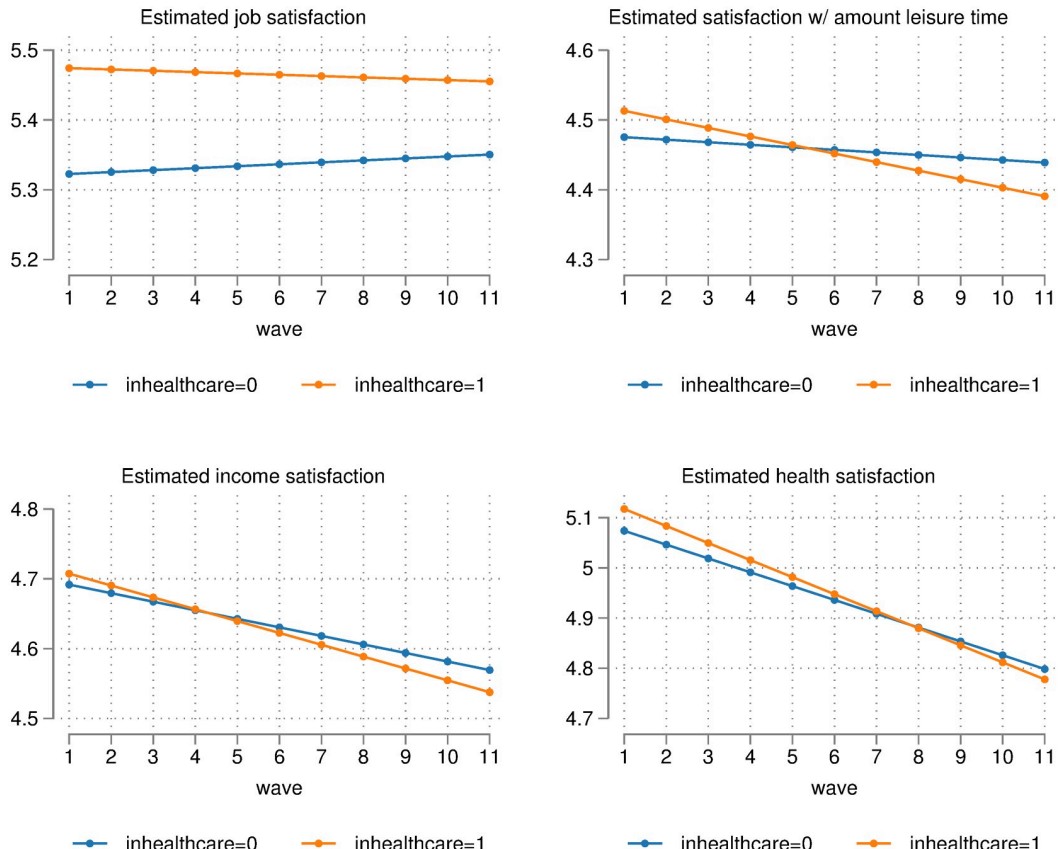

**Fig 2. Difference in satisfaction levels of healthcare workers over time, relative to workers outside healthcare (plotted using FE regressions shown in Table 1).**

does suggest that the job satisfaction gap between healthcare workers and non-healthcare workers may be slowly closing, but this trend is not significant. Regression (3) shows that there is also a decreasing trend for income satisfaction, though this is not statistically significant. Regression (4) has significantly negative time trends but non-significant 'in healthcare' dummies and 'in healthcare' × wave interaction terms. This suggests that estimated health satisfaction has decreased over the past decade for workers in all sectors.

**3.2.1 Comparison with other public sector workers.** Table 1 regressions (5) to (8) repeat (1) to (4), but separates non-healthcare workers into public sector (excluding healthcare) and non-public sector (i.e. private sector). The reference group for these regressions is healthcare workers, so that we can compare relative differences in healthcare with other sectors. Estimated satisfaction levels for each sector are plotted in Fig 3. This highlights stark differences in satisfaction between private and public sector. The convergence in job satisfaction between healthcare workers and non-healthcare workers appears to have been driven by an increasing level of job satisfaction in the private sector, as can be seen by the significantly positive interaction term for non-public sector × wave in regression (5). Estimated job satisfaction in the public sector (excluding healthcare) has fallen significantly over the period relative to healthcare. This is supported by a significantly negative interaction term for public sector × wave. Estimated private sector job satisfaction by wave 11 is higher than estimated public sector (excluding healthcare) job satisfaction. Hence, while overall job satisfaction appears to have

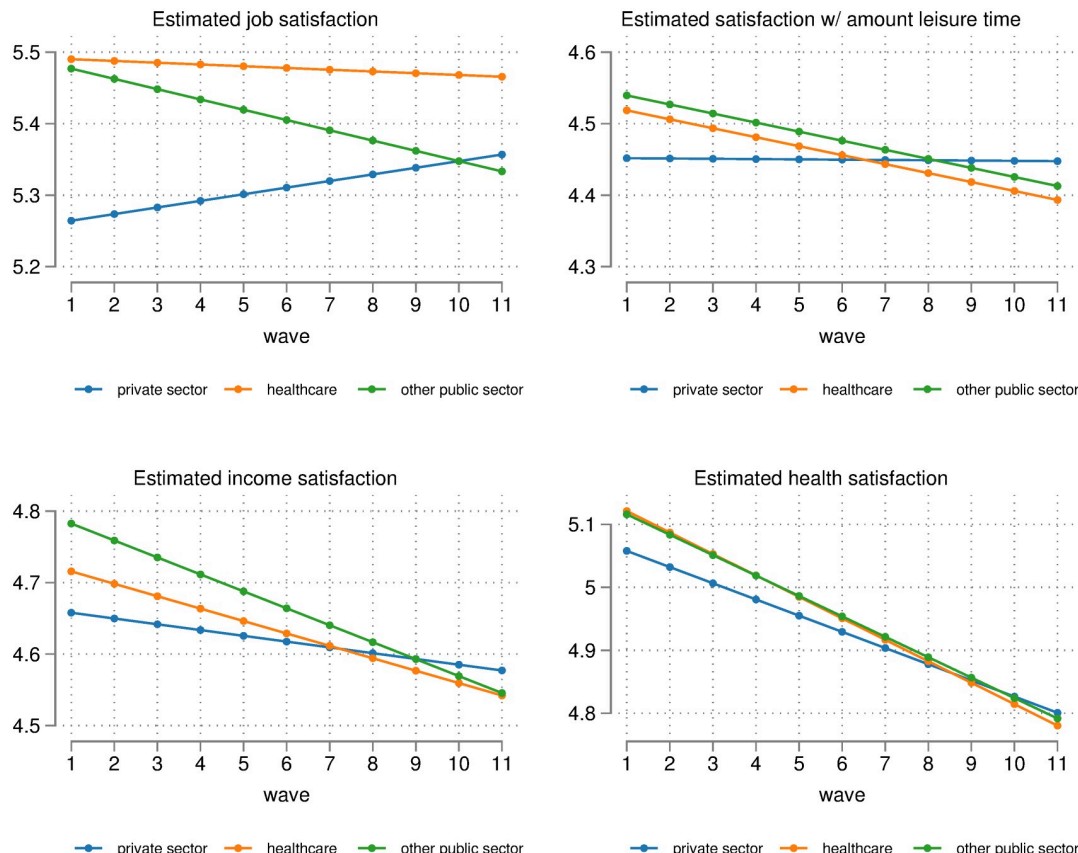

**Fig 3. Difference in satisfaction levels of healthcare workers over time, relative to other public sector workers and non-public sector workers (plotted using FE regressions shown in Table 1).**

deteriorated only slightly for healthcare workers, other parts of the public sector seem to have taken a larger hit.

Next, we see that the estimated rate of decline in satisfaction with the amount of leisure time is broadly the same between healthcare workers and other public sector workers, though the estimated level of satisfaction for is consistently lower for healthcare workers. While the drop in estimated satisfaction with the amount of leisure time is three times higher for healthcare vs non-healthcare workers (Fig 2), it is 31.25 times higher when comparing healthcare workers to non-public sector workers (Fig 3). This is because estimated satisfaction with the amount of leisure time has barely changed in the private sector over the decade. Again, the slight decline in satisfaction for non-healthcare workers in Fig 2 appears to have been driven by other public sector workers rather than workers outside the public sector.

Both estimated income satisfaction and estimated health satisfaction have declined for all workers over the period. However, the rate of decline has been slower for workers outside the public sector. Estimated income satisfaction was highest for non-healthcare public sector workers at the beginning of the decade, but it is highest for private sector workers at the end of the decade. Estimated income satisfaction of non-healthcare public sector workers has declined the most rapidly, where it is virtually identical to the income satisfaction of healthcare workers by the end of the decade. In summary, regressions (5) to (8) show that estimated satisfaction levels for healthcare workers have deteriorated over the past decade, particularly when viewed in relation to estimated satisfaction levels of private sector employees. However,

workers elsewhere in the public sector have faced at least as large of a decline in satisfaction levels as healthcare workers.

We also repeated these regressions using the six different measures capturing job-related mood. The estimates are shown in S1 Table, and trajectories for mood for workers across different sectors over the survey period are shown in S1 Fig. Overall, these estimates show that all workers have suffered from a decline in job-related moods over the study period. Healthcare workers have not experienced a significantly greater rate of worsening mood over the period relative to private sector workers. However, the same is not true for other public sector workers, who have experienced a significantly greater decline in job-related moods relative to private sector workers. This result stands for all mood variables apart from 'gloomy' after applying Bonferroni corrections for six hypotheses (given that we test six different measures of mood).

### 3.3 Predicting healthcare employment / attrition using lagged satisfaction

We next estimate models that predict whether a worker has remained in or joined the healthcare sector. From our general regression specification, we are now interested in the estimate of $\phi$, because this captures how important different aspects of satisfaction in one wave are in determining the likelihood of a person working in healthcare in the following wave. A significantly large $\phi$ indicates that the predictive power of satisfaction on the likelihood of joining / remaining in the healthcare sector *changes over time*. Given that previous literature and NHS workforce data suggest a worsening work-life balance as being a major reason behind NHS job attrition, we would expect that low satisfaction levels (particularly with the amount of leisure time) would predict an ever-lower likelihood of joining / remaining healthcare as we progress through the previous decade.

The first three fixed-effects OLS regressions in Table 2 are estimated for people who had been working in healthcare in at least one wave of the survey. Regression (1) in Table 2 estimates the likelihood of being in healthcare work in the current wave. We use regular OLS standard errors, as individuals were sampled randomly from the population (see [76]). We did also compute standard errors clustered by person, though these did not differ substantially from traditional OLS errors. For our set of contemporaneous covariates (i.e. $X_{i,t}$ in our specification), we included the cumulative number of waves that the individual had spent working in healthcare to date (as this would clearly affect the likelihood of remaining in a healthcare job), as well as the standard demographic variables from the previous analysis. Regression (1) shows that all four lagged satisfaction × wave interaction term coefficients are statistically significant at least at the 5% level. Health satisfaction and satisfaction with the amount of leisure have positive wave interaction terms, whereas income satisfaction and overall job satisfaction have negative wave interactions. This suggests that *lower* health or leisure time satisfaction has become increasingly predictive of leaving / staying out of healthcare as time goes on, whereas *higher* income satisfaction or overall job satisfaction has become increasingly predictive over time of leaving / staying out of healthcare.

For regression (2) in Table 2, we replaced job satisfaction with individual job mood variables, which were only collected in even numbered waves within the panel. This improves overall model fit ($R^2$ increases from 0.075 to 0.161). The four main lagged satisfaction × wave interaction terms increase in magnitude, adding further robustness to their predictive power. However, because mood variables were only available in even waves, the number of observations on which the model is fitted is smaller than in (1). Finally, regression (3) repeats (1) but adds contemporaneous satisfaction variables. This lowers the magnitudes of the lagged satisfaction × wave interaction estimates slightly, reducing the significance levels of the lagged

**Table 2. Fixed-effects OLS regressions predicting whether individuals were working in healthcare from current and lagged variables.**

| | Dependent variable: in healthcare (binary variable) | | | | | |
|---|---|---|---|---|---|---|
| | Only people that ever worked in healthcare | | | All employed people | | |
| | (1) | (2) | (3) | (4) | (5) | (6) |
| Cumulative yrs in health | 0.0427*** | 0.0260*** | 0.0440*** | 0.0143*** | 0.00769*** | 0.0144*** |
| | (0.00296) | (0.00410) | (0.00297) | (0.000502) | (0.000695) | (0.000502) |
| Age | 0.0192 | 0.00306 | 0.02 | 0.00145 | 0.000125 | 0.00145 |
| | (0.0132) | (0.0191) | (0.0132) | (0.00124) | (0.00226) | (0.00124) |
| Age^2 | -0.000323*** | -0.000212*** | -0.000333*** | -2.58e-05*** | -1.70e-05** | -2.57e-05*** |
| | (5.62e-05) | (7.72e-05) | (5.63e-05) | (5.83e-06) | (8.17e-06) | (5.85e-06) |
| Married | -0.0137 | 0.0022 | -0.0125 | -0.00178 | 0.0000708 | -0.00164 |
| | (0.0150) | (0.0218) | (0.0150) | (0.00167) | (0.00242) | (0.00167) |
| Num dependent children | -0.00392 | -0.0198** | -0.00384 | 0.0000966 | -0.00196* | 0.000117 |
| | (0.00688) | (0.00970) | (0.00688) | (0.000766) | (0.00110) | (0.000767) |
| Has degree | -0.0572* | 0.00464 | -0.0576* | -0.0132*** | 0.00157 | -0.0131*** |
| | (0.0310) | (0.0467) | (0.0310) | (0.00491) | (0.00719) | (0.00491) |
| Monthly income | 0.00000415 | 1.40e-05** | 0.00000446 | 0.000000417 | 8.96e-07* | 0.000000446 |
| | (3.85e-06) | (5.48e-06) | (3.86e-06) | (3.20e-07) | (4.69e-07) | (3.21e-07) |
| In salaried emp (vs self emp) | 0.344*** | 0.350*** | 0.346*** | 0.0210*** | 0.0197*** | 0.0213*** |
| | (0.0220) | (0.0334) | (0.0220) | (0.00178) | (0.00263) | (0.00178) |
| *Lagged variables* | | | | | | |
| In healthcare (t-1) | 0.151*** | 0.318*** | 0.149*** | 0.179*** | 0.339*** | 0.179*** |
| | (0.00832) | (0.0123) | (0.00832) | (0.00269) | (0.00402) | (0.00269) |
| Health satisfaction (t-1) | -0.00402 | -0.0122 | -0.00258 | -0.000551 | -0.00142 | -0.000432 |
| | (0.00458) | (0.00765) | (0.00467) | (0.000528) | (0.000874) | (0.000536) |
| Wave x health sat (t-1) | 0.00131* | 0.00235** | 0.00109 | 0.000172** | 0.000287** | 0.000157* |
| | (0.000687) | (0.00105) | (0.000709) | (7.99e-05) | (0.000121) | (8.18e-05) |
| Income satisfaction (t-1) | 0.0192*** | 0.0265*** | 0.0153*** | 0.00216*** | 0.00296*** | 0.00175*** |
| | (0.00526) | (0.00856) | (0.00543) | (0.000599) | (0.000973) | (0.000616) |
| Wave x income sat (t-1) | -0.00404*** | -0.00545*** | -0.00332*** | -0.000454*** | -0.000602*** | -0.000380*** |
| | (0.000779) | (0.00118) | (0.000817) | (8.90e-05) | (0.000133) | (9.28e-05) |
| Sat w/ amount leisure (t-1) | -0.0184*** | -0.0196** | -0.0161*** | -0.00196*** | -0.00226** | -0.00171*** |
| | (0.00532) | (0.00867) | (0.00544) | (0.000579) | (0.000947) | (0.000591) |
| Wave x sat w/ amount leisure (t-1) | 0.00326*** | 0.00387*** | 0.00279*** | 0.000338*** | 0.000419*** | 0.000288*** |
| | (0.000779) | (0.00118) | (0.000804) | (8.54e-05) | (0.000129) | (8.80e-05) |
| Job satisfaction (t-1) | 0.00701** | | 0.00679* | 0.000359 | | 0.000349 |
| | (0.00345) | | (0.00349) | (0.000357) | | (0.000360) |
| Wave x job sat (t-1) | -0.00146*** | | -0.00142*** | -0.000102** | | -0.000103** |
| | (0.000477) | | (0.000483) | (4.96e-05) | | (5.02e-05) |
| Job feel tense (t-1) | | -0.0260* | | | -0.00297* | |
| | | (0.0155) | | | (0.00179) | |
| Wave x job feel tense (t-1) | | 0.00470** | | | 0.000614** | |
| | | (0.00212) | | | (0.000245) | |
| Job feel uneasy (t-1) | | -0.00154 | | | 0.000535 | |
| | | (0.0180) | | | (0.00206) | |
| Wave x job feel uneasy (t-1) | | 0.000231 | | | -0.0000905 | |
| | | (0.00243) | | | (0.000283) | |
| Job feel worried (t-1) | | 0.0112 | | | 0.00106 | |
| | | (0.0175) | | | (0.00198) | |

(*Continued*)

**Table 2.** (Continued)

| | Dependent variable: in healthcare (binary variable) | | | | | |
|---|---|---|---|---|---|---|
| | Only people that ever worked in healthcare | | | All employed people | | |
| | **(1)** | **(2)** | **(3)** | **(4)** | **(5)** | **(6)** |
| Wave x job feel worried (t-1) | | -0.00424* | | | -0.000510* | |
| | | (0.00237) | | | (0.000270) | |
| Job feel depressed (t-1) | | -0.00466 | | | 0.000182 | |
| | | (0.0212) | | | (0.00240) | |
| Wave x job feel depressed (t-1) | | 0.00373 | | | 0.000309 | |
| | | (0.00288) | | | (0.000327) | |
| Job feel gloomy (t-1) | | 0.0406* | | | 0.00399 | |
| | | (0.0219) | | | (0.00253) | |
| Wave x job feel gloomy (t-1) | | -0.00418 | | | -0.000376 | |
| | | (0.00302) | | | (0.000350) | |
| Job feel miserable (t-1) | | -0.0291 | | | -0.0037 | |
| | | (0.0221) | | | (0.00251) | |
| Wave x job feel miserable (t-1) | | -0.000133 | | | 0.000065 | |
| | | (0.00307) | | | (0.000344) | |
| *Includes satisfaction vars at time t?* | *No* | *No* | **Yes** | *No* | *No* | **Yes** |
| Wave | -0.0076 | -0.0046 | -0.0064 | 0.0006 | 0.000285 | 0.000885 |
| | (0.0128) | (0.0184) | (0.0137) | (0.00121) | (0.00217) | (0.00132) |
| Constant | 0.0112 | 0.38 | -0.0797 | 0.0295 | 0.0583 | 0.0209 |
| | (0.483) | (0.693) | (0.483) | (0.0449) | (0.0820) | (0.0451) |
| Observations | 17709 | 9125 | 17709 | 165607 | 85153 | 165607 |
| R-squared | 0.075 | 0.161 | 0.078 | 0.045 | 0.129 | 0.046 |
| Number of individuals | 3533 | 3228 | 3533 | 36364 | 32114 | 36364 |

Note: Standard errors are in parentheses.

*** $p < 0.01$,

** $p < 0.05$,

* $p < 0.1$.

Regressions are for within-person longitudinal fixed-effects OLS models. For the first 3 regressions, the sample is restricted to people who worked in healthcare for at least one wave.

overall job satisfaction and lagged health satisfaction interactions. However, notably lagged satisfaction with the amount of leisure time remains a strong predictor of an ever lower likelihood of joining / remaining in healthcare over time. For additional robustness, we estimated three further models, (4) to (6), that repeat (1) to (3) but on the full sample. This means that the sample for this model included some people that had never worked in healthcare in any wave. Though parameter estimates are slightly different, the conclusions do not qualitatively change, aside from the fact that the predictive power of wave × lagged overall job satisfaction becomes negligible.

Fig 4 plots the predicted probability of working in healthcare across the waves of the UKHLS for different levels of satisfaction. The predictions are based on estimates from (1) in Table 2. Each panel in Fig 4 shows the moderating impact of a specific lagged satisfaction variable, and how this moderating effect changes over time, holding all other variables constant at their means. The satisfaction variables are measured on a 1–7 scale, however, we have chosen to show only the minimum, maximum, and midpoint of the scale for clarity. There are

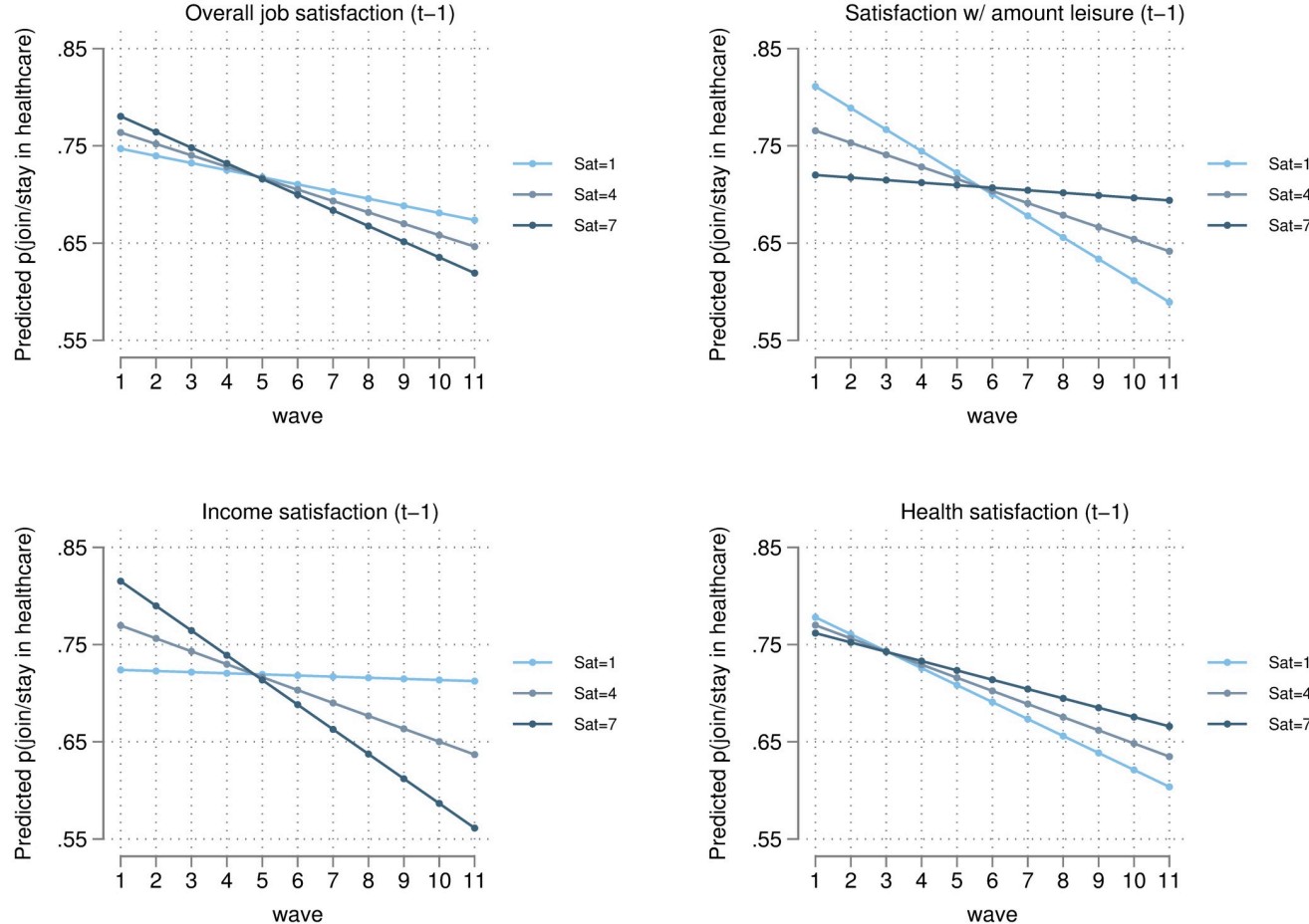

**Fig 4. Predicted probability of joining/staying in the healthcare sector with different levels of lagged satisfaction, measured on 1–7 scale (estimated from FE regression 1 in Table 2).**

particularly strong moderating effects for lagged satisfaction with the amount of leisure time and lagged income satisfaction. If we look only at healthcare workers in t-1, the predicted probability that they would stay in healthcare in period t if their income satisfaction was at its minimum in t-1 (i.e. 1) is 0.772 in wave 1 and 0.761 in wave 11. However, the predicted probabilities if their income satisfaction was at its maximum are 0.863 in wave 1 and 0.609 in wave 11. Therefore, healthcare workers with low income satisfaction were equally likely to stay in healthcare across the decade, whereas healthcare workers with high income satisfaction became considerably less likely to stay in healthcare over the decade.

The predicted probability of working in healthcare for people with the highest satisfaction with the amount of leisure time they had in the previous time period (satisfaction = 7) was approximately 0.720 in wave 1 and 0.694 in wave 11, i.e. about a 3 percentage point drop over the decade. In comparison, for people who were the least satisfied with the amount of leisure time they had (satisfaction = 1), the predicted probability of working in healthcare in the following wave was 0.811 in wave 1 and 0.589 in wave 11, i.e. about a 22 percentage point drop in the same timeframe. If we repeat this exercise only for individuals who were already working in healthcare (i.e. lagged 'in healthcare' = 1), then the likelihood of a healthcare worker remaining in healthcare in the next year when they have the maximum satisfaction with their amount

of leisure time drops from 0.768 in wave 1 to 0.742 in wave 11. However, were they to have the minimum level of satisfaction with their amount of leisure, the likelihood of them staying in healthcare falls from 0.859 in wave 1 to 0.637 in wave 11. Therefore, healthcare workers who were dissatisfied with the amount of leisure time they had were increasingly less likely to remain in the healthcare sector the closer we approach the present.

## 4 Discussion

In summary, our analysis of NHS workforce data shows that although the overall number of leavers has remained fairly constant over the past decade from Q1 2011 to Q3 2021, the proportion of leavers resigning voluntarily has steadily increased over the period. Poor work-life balance is the most rapidly increasing voluntary reason for leaving the NHS, and has become the most important explicitly specified voluntary reason for leaving overall. Building upon this, fixed-effects estimates from UKHLS data show that the average healthcare worker has seen a sharper reduction in satisfaction with their amount of leisure time over the past decade than the average non-healthcare worker. Over the previous decade, the proportion of NHS leavers that left for work-life balance reasons has approximately tripled. This appears to be consistent with our UKHLS estimates, which suggest that there is an estimated tripling of the drop in satisfaction with the amount of leisure time among healthcare workers relative to workers outside healthcare. Although satisfaction levels appear to have declined for all workers in general over the decade, estimated job satisfaction has risen for private sector workers whereas it has fallen in the public sector. Lastly, job-related satisfaction measures are significant predictors of the likelihood of joining or staying in healthcare in the future. Over the past decade, workers that have *relatively low* levels of satisfaction with the amount of their leisure time have become less and less likely (compared with those that are satisfied with their amount of leisure) to either join or remain in the healthcare sector in the following year. This is also true for workers who have *relatively high* levels of income satisfaction.

Although overall job satisfaction within healthcare remains relatively high compared with other sectors, the gap is shrinking. We think that our findings paint a worrying picture for the UK healthcare sector, and seem to echo findings from previous smaller-sample qualitative research studies that highlight an increasing level of concern about workload and stress. The increased disparity in job-related satisfaction between workers inside and outside of the healthcare sector seems to suggest that we will continue to see people leaving the sector in order to improve their levels of well-being, while people remaining in the sector experience deteriorating levels of well-being relative to their peers outside the sector. Our analyses also highlight that rapidly deteriorating satisfaction levels may be endemic within the UK public sector more generally. Although the exact reasons for this may differ across different parts of the public sector, one culprit may be the prolonged effects of austerity policies that reduced government spending and increased taxation near the beginning of the sample period [77].

The main limitations with our study stem from the characteristics of the data that we had access to. The reasons for leaving a job reported in NHS workforce statistics are only available at an aggregated level, presumably for anonymity reasons. Therefore, it was not possible to isolate whether there were more issues with work-life balance in specific parts of the healthcare sector. Additionally, as we explained in the methods section, NHS headcounts only cover employees that are directly paid by the NHS via the Electronic Staff Record system. Crucially, this does not include GPs or dentists, since they are considered to be independent contractors by the NHS. GPs and dentists may have been part of the UKHLS sample, though the occupational codes provided in the dataset do not allow us to break down health professionals into sub-categories. In the panel data analyses, though we identified satisfaction with the amount of

leisure time as being highly relevant to both potential attrition and as having deteriorated over time more rapidly in healthcare workers, we were unable to isolate forms of satisfaction that were more specific about the exact type of poor working conditions that workers might have been suffering from. Furthermore, although we were able to use a measure of satisfaction with the *amount* of leisure time, we would also have liked to see some measure of the *quality* of leisure time experienced by healthcare workers. While this information was collected in the older British Household Panel Survey, it appears to not be available in the UKHLS. We think the inclusion of leisure quality as a measure is important in future longitudinal panel surveys, because it is likely that this information would be strongly related to various different subjective well-being outcomes. It may also be able to highlight relatively undesirable working conditions in an industry or occupation.

Our findings are also unable to shed light on any differences in healthcare worker satisfaction across devolved UK nations. NHS workforce data only covers England, and the UKHLS sample contains too few healthcare workers outside England to make robust comparisons across the four NHS systems within the UK. It is generally difficult to compare conditions across nations because they do not collect the same statistics [78]. However, there are on average approximately 34% more nurses and 18% more GPs per 1000 patients in the three nations outside of England, with much of the difference driven by Northern Ireland and Scotland in particular. There do seem to be cultural differences in approach to healthcare between nations. For example, Scotland has taken objection to the 'internal competition' approach employed within NHS trusts in England, and has made the most progress towards better workforce planning [78, 79]. However, more work (along with consistent statistical reporting) is necessary to understand whether this translates to different satisfaction levels across the devolved nations.

We see our study as complement to smaller scale but more targeted studies that have already reported recent first-hand accounts of poor NHS worker experiences. For example, one NHS frontline junior doctor in England during the first wave of COVID-19 in the UK highlighted a lack of confidence in government planning during the pandemic, personal burnout leading to the formation of unhealthy dietary and exercise habits, and a deterioration in sleep quality [80]. While the pandemic exacerbated poor working conditions for NHS staff, concerns about governance and workload within the NHS have existed prior to this. GPs for example have been driven towards meeting bureaucratic targets in order to secure funding from commissioning groups, at the expense of both patient care and doctors' well-being [48]. The extent of these poor conditions is, in some cases, shocking. For example, one doctor wondered whether they would have enough time in the day to go to the toilet [48]. Most people would regard such working conditions as unacceptable. Yet, healthcare workers face such conditions while also being responsible for the health and well-being of others.

What can be done about the decline in healthcare working conditions? There appear to be two main approaches from a policy perspective. The first approach is top-down–structural changes to the UK healthcare system that protect staff well-being while incentivising patient care. Excessive workload and insufficient leisure time is, at the simplest level, a sign of a resource shortage. A 2021 NHS staff survey completed by over 600,000 workers revealed that only 27% thought staffing was at a sufficient level in their specific organisation, a reduction of 11 percentage points from 2020 [81]. It appears as though at least part of the reason behind poor staffing levels is political inaction on workplace planning reforms [82]. If staff are leaving at an accelerated rate or are hesitant to join, a basic labour supply model would indicate that the wage is too low. Increasing wages may help to attract or retain healthcare workers to some extent, though this may not fully compensate for adverse working conditions if outside options are still more attractive. It is also possible that an uncertain environment within the NHS is partly to blame for attrition. For example, GPs that have moved abroad after training in the

UK tend to be more risk averse [83]. This suggests that any structural reform (independent of wage increases) would likely also need to make working conditions more stable and predictable if the NHS wants to retain talent. A comprehensive 2021 Lancet report has already listed a number of recommendations for structural reform within NHS [79]. These recommendations echo a need for increased funding (4% per year in real terms, funded by tax increases), better workforce planning/strategy, and notably also the removal of competition requirements in England [79].

The second approach is bottom-up–a change in working culture coupled with provision of workplace measures that improve the day-to-day experiences of workers in the NHS. Smaller-scale well-being policies may be able to alleviate stress at a more localised level. For example, Supported Well-being Centres were set up following the COVID-19 pandemic, which provided relaxing spaces as well as well-being 'buddies' for emotional/psychological support [84]. These measures were highly valued, but are still quite resource intensive. Less resource intensive interventions could also include, for example, providing vouchers and incentives for leisure activities. However, if the problem is largely caused by a lack of leisure time, then these interventions are likely to be ineffective. We emphasise that we do not think only one approach is sufficient to solve the problem of deteriorating conditions or work-life balance. Rather, it seems prudent to pursue longer-term structural changes with the support of shorter-run (and less costly) well-being interventions.

## 5 Conclusion

This study quantitatively investigated whether working conditions in the NHS have deteriorated significantly over the past decade. Our results suggest that while overall job satisfaction in healthcare has been relatively high, there is significant evidence that satisfaction levels have declined over the past decade, particularly in terms of work-life balance and leisure time. Work-life balance accounted for almost three times as many NHS job leavers in 2021 than it did in 2011. Low satisfaction with the amount of one's leisure time has become increasingly more predictive of a future outside of the healthcare sector as the past decade has progressed. Additionally, satisfaction with amount of leisure time for healthcare workers has fallen by three times the amount that it has fallen for non-healthcare workers between 2010 and 2020.

Our study has implications for both policy and additional research. Our findings support a need for structural change within the UK healthcare sector to improve working conditions. This seems necessary both to attract and to retain staff, given the increasing income and leisure time disparity between healthcare work and employment outside the sector. There may also be scope for smaller-scale well-being interventions at the 'ground level' that could provide temporary respite to healthcare workers. Future work on this topic should focus on identifying specific causes for the deterioration in work-life balance, understanding how working conditions in healthcare differ across the four devolved UK NHS systems, and testing workplace interventions that may boost well-being and retention in the short-term as larger-scale policy changes are being implemented in the longer-term.

## Supporting information

**S1 Table. How the trajectory of job related feelings over time differs in healthcare workers, relative to other public and private sector workers (fixed-effects OLS models).**
(PDF)

**S1 Fig. Estimated job related feelings of people in the healthcare sector over the past decade, relative to other public sector workers, and private sector workers.**
(PDF)

## Acknowledgments

The authors thank Andrew Oswald, Thomas Hills, and Mikhail Spektor for their helpful comments.

## Author Contributions

**Conceptualization:** Neel Ocean.

**Data curation:** Neel Ocean.

**Formal analysis:** Neel Ocean.

**Investigation:** Neel Ocean.

**Methodology:** Neel Ocean.

**Software:** Neel Ocean.

**Visualization:** Neel Ocean.

**Writing – original draft:** Neel Ocean.

**Writing – review & editing:** Neel Ocean, Caroline Meyer.

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
