## [Decision Letter · Decision Letter 0]

17 Feb 2023

PONE-D-22-34427

Satisfaction and attrition in the UK healthcare sector over the past decade

PLOS ONE

Dear Dr. Ocean,

Thank you for submitting your manuscript to PLOS ONE. After careful consideration, we feel that it has merit but does not fully meet PLOS ONE’s publication criteria as it currently stands. Therefore, we invite you to submit a revised version of the manuscript that addresses the points raised during the review process.

Please ensure that you address the three minor issues raised by the reviewer specifically on the necessity of 'systemic changes' and a discussion on devolution and its implications.

We look forward to receiving your revised manuscript.

Kind regards,

Nafis Faizi, MD, MPH

Academic Editor

PLOS ONE

Journal Requirements:

Reviewers' comments:

Reviewer's Responses to Questions

**Comments to the Author**

1. Is the manuscript technically sound, and do the data support the conclusions?

Reviewer #1: Yes

2. Has the statistical analysis been performed appropriately and rigorously? 

Reviewer #1: Yes

3. Have the authors made all data underlying the findings in their manuscript fully available?

Reviewer #1: Yes

4. Is the manuscript presented in an intelligible fashion and written in standard English?

Reviewer #1: Yes

5. Review Comments to the Author

Reviewer #1: Review for PLoS ONE of satisfaction and attrition in the UK healthcare sector

This is a really nice article on an important issue, well executed, with strong and important findings, and I see no reason not to publish it. It starts with a well-executed literature review on the topic of working conditions and their effects in the "UK NHS." They find a number of alarming trends in particular elements of job satisfaction which seem to explain departures from the NHS. Alarmingly, both the reasons for dissatisfaction and the departures are going up. COVID-19, austerity, and the HSCA all seem to play a role.

Three minor points:

The policy recommendations are in a sense obvious, but it is not clear why "systemic changes" are necessary. The authors seem to make a case that more money, more staff, and better management are what the NHS systems need (as they say, the problems they report are evidence of a resource shortage). The fact that they report an uncertain environment is a cause of departures seems to suggest policymakers ought to avoid unspecified systemic change.

There is a case for at least some discussion of devolution, which in principle might lead to variation in quality of management, uncertainty, public sector pay, and such. I don't know of the surveys have the right samples to say anything illuminating but the authors might at least discuss the issue. It seems to be the kind of issue for which there is scope to learn useful things across the four UK systems.

Otherwise, it's paltry, but I would recommend the authors give it one more read to tune up the prose a bit. The abstract, first page, introduction to the literature review. and concluding sections can be a bit choppy and obscure the motivation and findings on the first or second reading.

6. PLOS authors have the option to publish the peer review history of their article (what does this mean?). If published, this will include your full peer review and any attached files.

Reviewer #1: No

---

## [Author Response · Author response to Decision Letter 0]

27 Feb 2023

Please see attached response to reviewers document with the revised submission. This outlines how changes were made to the paper in response to each of the reviewer's points.

---

## [Editor Report · Decision Letter 1]

3 Apr 2023

Satisfaction and attrition in the UK healthcare sector over the past decade

PONE-D-22-34427R1

Dear Dr. Ocean,

We’re pleased to inform you that your manuscript has been judged scientifically suitable for publication and will be formally accepted for publication once it meets all outstanding technical requirements.

Kind regards,

Nafis Faizi, MD, MPH

Academic Editor

PLOS ONE

Additional Editor Comments (optional): All the comments of the reviewer has been duly discussed.
---

## [Editor Report · Acceptance letter]

5 Apr 2023

PONE-D-22-34427R1 

Satisfaction and attrition in the UK healthcare sector over the past decade 

Dear Dr. Ocean:

I'm pleased to inform you that your manuscript has been deemed suitable for publication in PLOS ONE. Congratulations! Your manuscript is now with our production department. 

Kind regards, 

on behalf of

Dr. Nafis Faizi 

Academic Editor

PLOS ONE